# Pre-Test Manipulation by Intraperitoneal Saline Injection with or without Isoflurane Pre-Treatment Does Not Influence the Outcome of Social Test in Male Mice

**Flóra Pomogyi** [1,2], **Bibiána Török** [1,2] **and Dóra Zelena** [1,2,*]

1  Institute of Physiology, Medical School, University of Pécs, Centre for Neuroscience, Szentágothai Research Centre, 7624 Pécs, Hungary
2  Laboratory of Behavioural and Stress Studies, Institute of Experimental Medicine, 1083 Budapest, Hungary
*  Correspondence: dora.zelena@aok.pte.hu

**Abstract:** Preclinical studies on rodents should follow the 3R principle minimising the suffering of the animals. To do so, some researchers use inhalation anaesthetic induction even before intraperitoneal injection. However, several studies suggested that both interventions might influence the behaviour of the animals. We aimed to test whether intraperitoneal injection alone or in combination with isoflurane anaesthesia is a preferable treatment method 30 min before a social test. Male C57BL/6 mice were studied using a behavioural test battery comparing three groups (one control group and intraperitoneal saline-treated groups with or without short isoflurane inhalation). Our results confirmed that both interventions had no profound influence on the conventionally measured parameters of social tests (interest in sociability, social discrimination memory, social interaction as well as resident–intruder test) and were not acutely stressful (measured by similar ACTH levels between the groups) not even after repeated administration (similar body weight gain during the one-week observation period). Taking into consideration the possible long-term harmful effect of isoflurane inhalation, we recommend using intraperitoneal injection without it as saline injection did not violate the 3R principle inducing only mild stress.

**Keywords:** intraperitoneal saline injection; isoflurane; sociability; social interaction; resident–intruder test

## 1. Introduction

Preclinical studies involving research animals remain a key component of discovery biology, as they are indispensable for the understanding of the corroborative biological processes as well as for identifying new treatment options and testing efficacy in disease models [1]. There are more than 20 million rodents (rats and mice) used in research annually [2]. To perform more humane animal experimentation the principles of the 3Rs (Replacement, Reduction and Refinement) were introduced over 50 years ago [1,3]. This principle sets out to tackle the fact that research very often causes pain. Although numerous methods are used to assess the presence or absence of pain, all of them are rather subjective [4].

For pain-induced stress, measurement of stress-hormone levels is a common method to assess the animal welfare [5]. Interestingly, activation of the hypothalamic–pituitary–adrenocortical axis (HPA, the so-called stress axis) has an analgesic effect, trying to counterbalance the source of the stressor in the organism [6]. Nevertheless, as glucocorticoids are the end hormones of the HPA axis, their levels (mainly corticosterone in rodents) are the most commonly studied [7]. As these steroid molecules are rather stable, and accumulating also in the hair/fur/feathers [8,9], as well as appearing in faecal samples [10,11], their measurement may provide information about long-term, chronic stressors. However, during acute stress situations their levels reaches a plateau rather early, and therefore they

cannot be used to judge the intensity of the stressor [12,13]. The pituitary component of the HPA axis, the adrenocorticotropin (ACTH), is known to be a better marker; thus, we concentrated on this biomarker.

As research moves towards better handling of the test subjects, it has to be considered whether short interventions, like intraperitoneal injections, can also be executed without causing any stress. Other than the humane aspects, reducing stress endured by the subjects would possibly have less influence on the behaviour of the animals, providing results that are more reflective of their behaviour under natural circumstances. Indeed, behavioural flexibility [14] as well as social interactions [15] are rather sensitive to stress. During social contact, stressful experience can even be transferred to other individuals [16]. Nevertheless, social contacts are of the utmost important not only for humans, but also for rodents [17]. Social feelings may deeply influence cognition, emotions, behaviours and well-being [18]. Moreover, social problems are often detectable in many neurological and psychiatric disorders [19,20]. To have a comprehensive picture about social behaviour, a battery of tests, rather than a single test, should be used. This is also in accordance with the 3R principle [1,3], as repeated usage of animals reduces the required number and social interactions are part of the everyday life, thus, their repeated application is considered as eustress [21]. Therefore, we conducted sociability, social interaction (SIT) and resident–intruder (RIT) tests on the same animals [22,23].

Studying brain mechanisms often requires some treatment of the animals. Moreover, new methods, like chemogenetic one, also make pre-test injections necessary, which is often done 30 min before the start of the behavioural observation [22]. Previous studies in mice showed that social recognition is not influenced when the mice are anaesthetized with isoflurane 30 min before the test, with the aim to avoid the stress of the intraperitoneal injection [24]. Although anaesthetics are supposed to reduce manipulation-induced stress, the loss of consciousness per se might be also stressful. Indeed, ether, an inhaled anaesthetic used for long time, is a strong activator of the HPA axis [25,26]. Isoflurane, a more favourable substance, is a fluorinated ether with general anaesthetic and muscle relaxant activities. Although the exact mechanism of action has not been established, inhaled isoflurane appears to act on the lipid matrix of the neuronal cell membrane, which results in disruption of neuronal transmission. It is on the World Health Organization's List of Essential Medicines [27], and it is increasingly used for rodent anaesthesia [28]. However, its usage is not without any harm. In rodent models it can induce neurodegeneration in neonates [29] and aversive behaviour in adults [2]. In rats, anxiogenic behaviour was detected one week after isoflurane inhalation [30]. Moreover, it elevates ACTH levels 5 min after its introduction, but these levels normalise within 30 min [7]. We have to admit, that corticosterone requires a longer time to normalise. This could be the reason that some authors even use this inhaled anaesthetic as a stressor [31].

On the basis of all above mentioned findings, there is insufficient information to evaluate whether intraperitoneal injection with or without isoflurane inhalation are stressors that influence experimental results of social behaviours in laboratory mice. Therefore, we aimed to study whether intraperitoneal injection with or without isoflurane anaesthesia applied 30 min before a social test has any effect on the measured behavioural outcomes and plasma ACTH levels (for the details of the experiments see Figure 1).

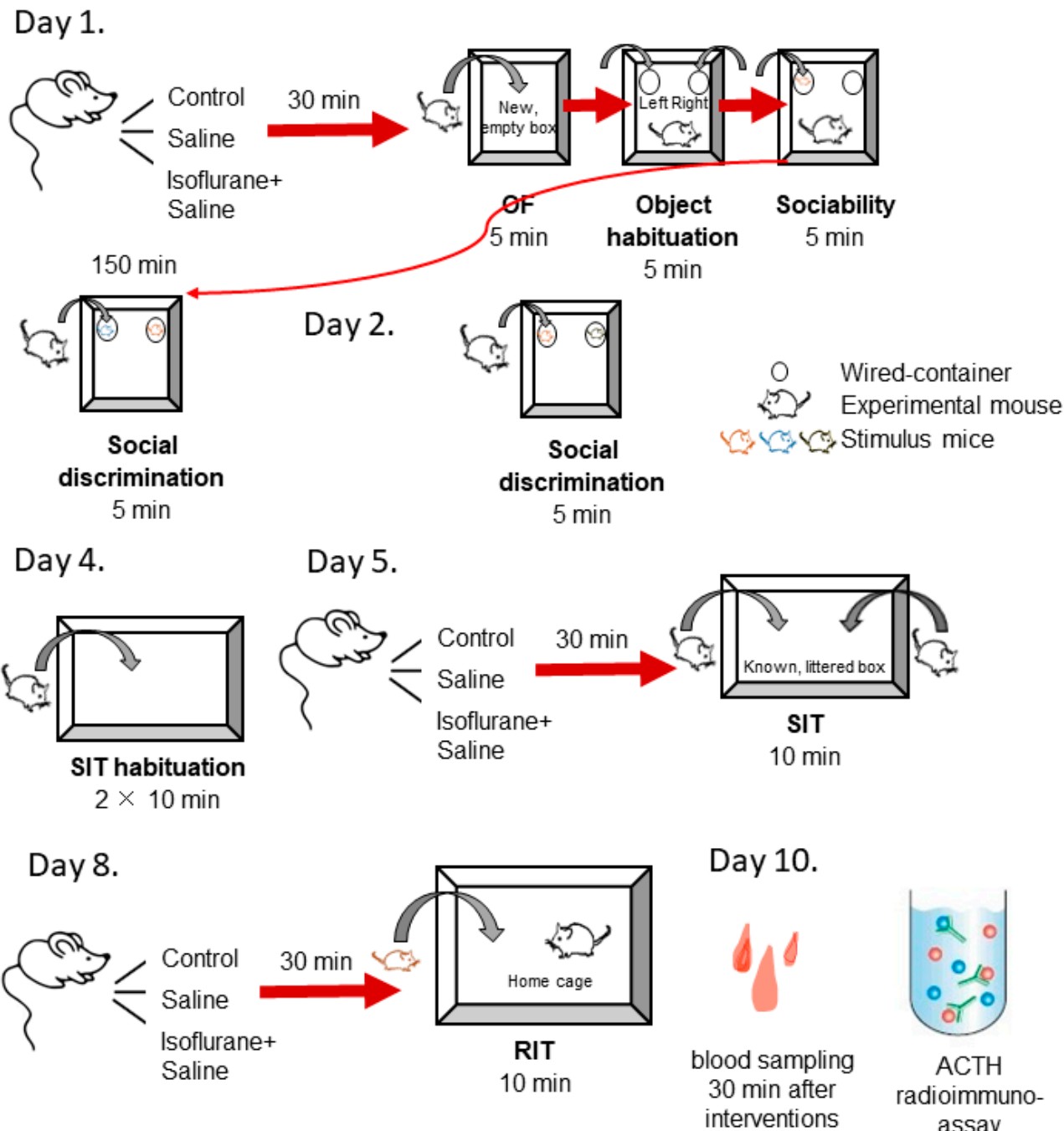

**Figure 1.** Timeline of the experiment. Three groups of C57Bl/6 male mice were compared: control, without any intervention (n = 11); saline, injected intraperitoneally with physiological saline (1 mg/10 mL/kg) 30 min before behavioural testing (n = 10); isoflurane+saline, isoflurane inhalation until loss of consciousness followed by saline injection as in the previous group (n = 10). Abbreviations: OF: open field; SIT: social interaction; RIT: resident–intruder test.

## 2. Results

The mice were randomly divided into three groups, thus, there were no significant difference in their initial body weight (Table 1). After repeated testing and interventions, most mice were able to gain some weight, but there were no significant differences between the different treatment groups.

Serum ACTH levels measured 30 min after the 4th treatment did not show any differences between the treatment groups (Figure 2A). Similarly, the treatment did not

influence the locomotion measured during the first 5 min of the sociability test, i.e., the open field (OF) phase (Figure 2B).

**Table 1.** Summary of the results. Data were analysed by one-way ANOVA (degree of freedom (2, 28)). For object habituation and sociability, the effect of treatment is given based upon repeated measure ANOVA analysis. The preference for stimulus mice during sociability was highly significant ($F_{(1,28)}$ = 17.24; $p$ < 0.01) without any influence of the treatment. * $p$ < 0.05 vs. saline-treated; the significant difference is highlighted by red.

| | | | **Control** | **Saline** | **Isoflurane** | **F** | *p* |
|---|---|---|---|---|---|---|---|
| General data | Body weight (g) | Initial | 25.28 ± 0.33 | 25.53 ± 0.59 | 25.17 ± 0.47 | 0.22 | 0.80 |
| | | Weight gain | 0.05 ± 0.35 | 0.58 ± 0.72 | 0.27 ± 0.87 | 0.17 | 0.84 |
| | | ACTH (fmol/mL) | 178.38 ± 8.81 | 142.36 ± 22.30 | 163.63 ± 22.46 | 1.04 | 0.37 |
| Sociability | Open field | Distance | 1714.34 ± 96.88 | 1780.49 ± 82.21 | 1738.79 ± 76.48 | 0.15 | 0.86 |
| | Object habituation | Left% Right% | 8.1 ± 3.00 6.46 ± 1.72 | 5.78 ± 0.93 7.44 ± 1.40 | 7.07 ± 2.54 7.96 ± 2.18 | 0.08 | 0.92 |
| | Sociability | Object% Mice% SI | 7.25 ± 1.88 21.71 ± 5.28 69.90 ± 7.74 | 4.84 ± 1.19 15.01 ± 3.74 73.51 ± 7.55 | 8.09 ± 1.99 17.25 ± 5.51 61.08 ± 7.36 | 0.65 0.66 | 0.53 0.52 |
| | Social discrimination | 2.5 h Old% 2.5 h New% 2.5 h DI | 14.14 ± 2.95 11.60 ± 2.71 −6.81 ± 10.76 | 12.31 ± 3.25 15.76 ± 3.66 9.32 ± 11.02 | 14.03 ± 2.25 12.16 ± 2.70 −12.84 ± 8.74 | 0.05 1.26 | 0.95 0.30 |
| | | 24 h Old% 24 h New% 24 h DI | 12.68 ± 3.27 13.75 ± 2.88 −0.03 ± 14.42 | 12.42 ± 2.66 16.01 ± 2.54 17.44 ± 7.25 | 10.78 ± 1.56 11.41 ± 2.20 −3.74 ± 9.39 | 0.48 1.00 | 0.63 0.38 |
| Social interaction test (time% of 10 min) | Friendly social interaction | | 19.05 ± 2.50 | 19.61 ± 1.94 | 24.00 ± 2.88 | 1.20 | 0.32 |
| | Aggression | | 3.68 ± 1.71 | 7.74 ± 2.91 | 2.06 ± 0.83 | 2.11 | 0.14 |
| | Friendly index | | 65.93 ± 6.14 | 52.61 ± 6.07 | 79.11 ± 5.91 * | 4.81 | <span style="color:red">0.02</span> |
| Resident–intruder test (time% of 10 min) | Aggression | | 5.77 ± 4.76 | 7.49 ± 7.97 | 3.31 ± 3.78 | 1.21 | 0.32 |
| | Friendly social interaction | | 23.23 ± 10.34 | 23.42 ± 10.85 | 18.92 ± 11.59 | 0.51 | 0.61 |
| | Friendly index | | 74.75 ± 6.19 | 71.32 ± 5.28 | 70.63 ± 9.01 | 0.11 | 0.90 |

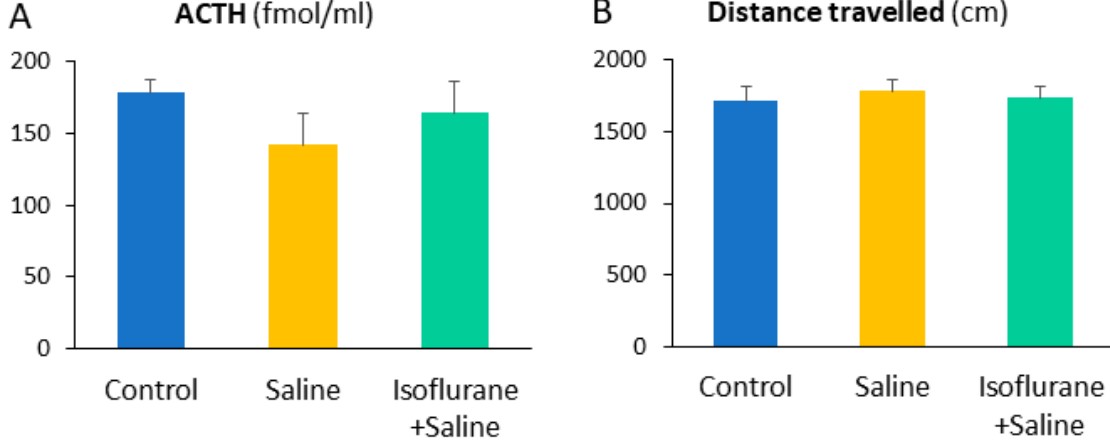

**Figure 2.** Serum ACTH (**A**) and locomotor (**B**) parameters 30 min after interventions. The comparison of the three groups of C57Bl/6 male mice did not reveal any differences (n = 10–11/group).

During the habituation phase of the sociability test, we could not detect any side preferences (Table 1), thus, the later measured social preferences were not confounded by this parameter. As expected, all the test mice were highly social, preferring the mice-containing object above the empty one. Although there was no direct difference between the groups, the sociability index (SI) significantly differed from the 50% chance level only in control and saline-injected mice, but not in isoflurane+saline-treated mice (control: t(10) = 2.57, *p* = 0.03; saline: t(9) = 3.12, *p* = 0.01; isoflurane+saline: t(8) = 1.50, *p* = 0.17) (Figure 3A).

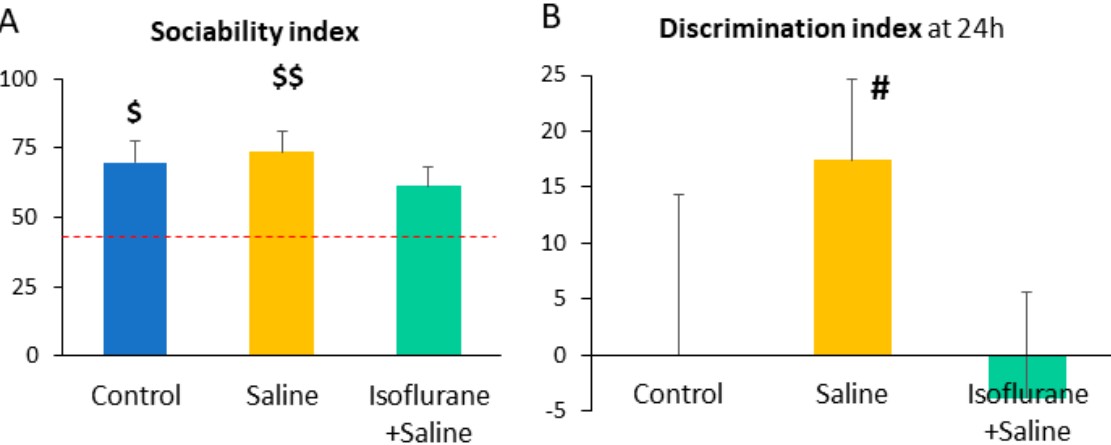

**Figure 3.** Sociability (**A**) and discrimination (**B**) indices. The comparison of the three groups of C57Bl/6 male mice did not reveal any differences (n = 8–10/group). However, only control and saline, but not the isoflurane+saline-treated group had intact social preference (**A**). At 24 h, social discrimination abilities were preserved only in the saline-treated group (**B**). \$ *p* < 0.05; \$\$ *p* = 0.01 vs. 50% chance level by single-sample *t*-test; # *p* < 0.05 vs. 0% chance level by single sample *t*-test.

During short term (2.5 h)- social memory testing, none of the groups showed the expected preference toward the new stimulus mice (control: t(8) = −0.63, *p* = 0.54; saline: t(8) = 0.85, *p* = 0.42; isoflurane+saline: t(8) = −1.47, *p* = 0.18) (Table 1). There was no difference between the groups either. However, during long-term (24 h) social memory testing, previously saline-injected animals were able to remember and spent significantly longer times investigating the new than the old stimulus mice reflected by significantly higher than 0 discrimination index (DI) levels (control: t(9) = −0.00, *p* = 1.00; saline: t(8) = 2.41, *p* = 0.04; isoflurane+saline: t(8) = −0.40, *p* = 0.70) (Figure 3B).

During the SIT conducted among highly anxious conditions (in light, new environment), there were no significant differences between the treatment groups in either studied parameter (friendly, aggressive or defensive behaviour) (Table 1). However, the friendly index (FI) parameter was highly significant among groups: the isoflurane+saline pretreated animals, whenever they showed any social behaviour, were more friendly than the saline-treated ones (Figure 4A).

During RIT when the animals are supposed to defend their own home, controversially, the friendly behaviour even increased compared to SIT (repeated measure ANOVA on FIs: F(1,25) = 4.30, *p* = 0.05) (Figure 4B). However, there were no differences in either studied parameter between the groups (Table 1).

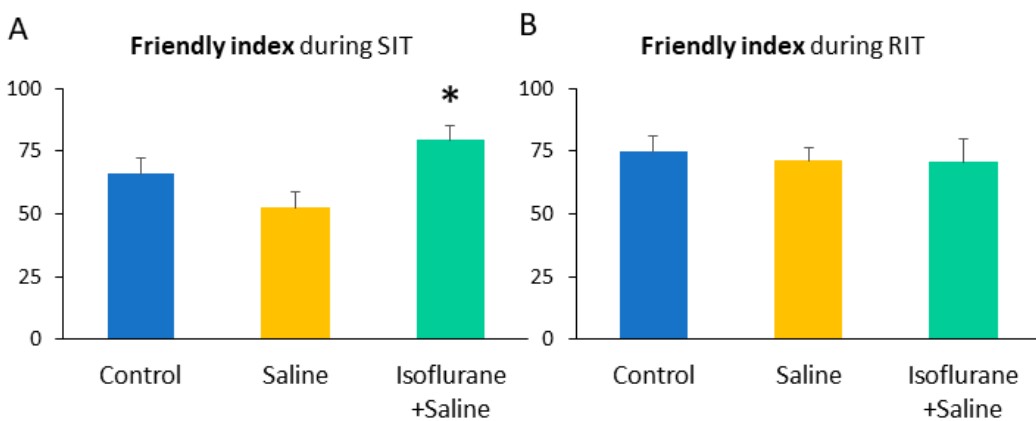

**Figure 4.** Friendly index during social interaction (SIT; **A**) and resident–intruder (RIT; **B**) tests. During SIT, the isoflurane+saline pre-treatment increased friendly social encounters among the social interactions compared to saline-treated mice (**A**). However, there was no difference between the groups during the RIT test (n = 9–10/group). * $p < 0.05$ vs. saline-treated group.

## 3. Discussion

Our results confirmed that intraperitoneal injection with or without isoflurane inhalation 30 min before social tests had no profound influence on the conventionally measured parameters of social tests and the mice were not acutely stressed (measured as similar ACTH levels between groups) or even after repeated administration (similar body weight gain during the 10-day observation period).

The body weight gain is typically decreased after application of chronic stressors not only in rats [32,33], but also in mice [34,35]. Thus, it was important to confirm that repeated interventions do not keep the animals in a constant stressed state. Moreover, the similar ACTH levels of the groups 30 min after our manipulation suggest that the effect of the acute stress of intraperitoneal injection or isoflurane inhalation+saline injection is transitory [7].

Locomotion is required for all kind of behaviour. Therefore, it is utmost important to confirm that our interventions had no effect on it, allowing proper interpretation of the results of the other tests. For the same reason, the lack of side preference had to also be confirmed. None of our intervention influenced these parameters. We have to admit, that our intervention was rather short (several min), while for anaesthesia purposes, longer treatment is needed. Indeed, a single 40 min isoflurane exposure already induced some motor coordination problems measured some days later and supported by alterations in the brain structure [36].

As for social behaviour, although major differences could not be observed, subtle effects were detectable. Unexpectedly, isoflurane pre-treatment made the animals less socially interested 40 min after its application (Figure 3A). It is important to note that even in this case, the test animals were sniffing the stimulus mice for 15–21% of the observation period, allowing enough time for recognition. Therefore, previous tests applying this anaesthetic to mice 30 min before social recognition, could not have detected any reduction in social memory [24]. Moreover, during previous experiments, isoflurane did not influence the social discrimination abilities of C57BL/6 mice, in accordance with the lack of late (up to 90 days) consequences on social memory after single (40 min) or repeated (twice, 3 days apart) inhalations [36]. Furthermore, one week after a single 2 h isoflurane anaesthetization, no effect on social interactions as well as on aggression in the tube test could be detected [37]. However, in the developing brain, a single 5–6 h or repeated 3 h isoflurane exposure might have detrimental late consequences, reducing the social contacts/discrimination as well as increasing anxiety (in Rhesus monkey: [38,39], in rats: [40], in mice: [41]). There might be some gender differences, with some authors reported male [42] or female rats [43] to be more vulnerable.

Interestingly, saline injection enhanced social memory formation that was measurable 24 h after its application (Figure 3B). We might assume that the discomfort of handling

and the small pain of the intraperitoneal injection is a mild stressor, which is thought to be beneficial for memory consolidation [44]. A mild, non-significant increase in the aggressive behaviour of intraperitoneally injected animals further supports their agitated, slightly stressed state, as stressors often precipitate violent behaviour [45]. Indeed, in accordance with the supposed pain- and thereby stress-reducing effect of isoflurane anaesthesia [28], mice were more friendly 30 min after its inhalation (Figure 4A).

To date, there is no information whether it is possible to reproduce this study's phenomena in other species, other strains, and females. A further limitation of our study is its focus on social behaviour.

## 4. Materials and Methods

### 4.1. Animals

Adult male mice (C57BL/6J, 14–15-week-old, the stimulus mice: 7–8-week-old) were bred at the Institute of Experimental Medicine, Budapest, Hungary, in Macrolon cages (40 cm × 25 cm × 26 cm, corn cob bedding) under a 12 h light–dark cycle, 21 ± 1 °C, 50–60% humidity, with food (standard mice chow, Charles River, Veszprém, Hungary) and tap water available ad libitum. The animals were kept 3–4/cage and were isolated 30 min before the first behavioural test to enhance social interest during subsequent tests. Behaviour was tested during the dark, active period (between 9:00 h and 13:00 h, lights on at 18:00 h) under dim light, except for SIT, during which lights were on.

### 4.2. Behavioural Testing

The animals were divided into three groups: 1. control, without any intervention before behavioural test (n = 11); 2. saline, injected intraperitoneally with 1 mL/kg physiological saline (26G needle, 1 mL syringe, <1 min immobilised and elevated by the skin of the back) at room temperature 30 min before starting the test (n = 10); 3. isoflurane+saline, animals were put in a jar containing isoflurane soaked cotton pad until loss of consciousness than injected intraperitoneally in similar manner as group 2 (n = 10).

The following test battery was conducted (Figure 1): day 1. 30 min after intervention: sociability; day 2. 24 h after intervention: social discrimination; day 4. SIT habituation, one mouse in a cage at a time, 2 × 10 min; day 5. SIT test 30 min after intervention; day 8. RIT 30 min after intervention; day 10. sacrifice for blood sampling 30 min after interventions by decapitation without further anaesthesia.

The behavioural experiments were recorded, and the videos were analysed later by Noldus Ethovision XT 15 (open field, OF, Wageningen, The Netherland), or computer-based event recorders (H77, Budapest, Hungary; Solomon Coder [https://solomon.andraspeter.com/ accessed on 1 January 2019]) by experimenters blind to the treatment.

#### 4.2.1. Sociability Test

It measures the interest of the tested animals towards conspecifics. The experimental room was dark, only lit by 20 lux infrared light. Mice were placed in empty, white plexiglass boxes covered by transparent plexiglass for 5 min (open field phase; OF). During this period the distance travelled was measured automatically. Right after this, two identical wired containers were placed into the test arena for 5 min and frequencies as well as time spent sniffing the two containers were recorded (habituation phase). Then, an unknown, juvenile conspecific was placed under one of the wired containers for 5 min. The animals could not contact physically, but were able to see, hear and smell each other (sociability phase). The sides of the stimulus animals (either under the right or left container) were interchanged between animals. Note that the sociability phase started 40 min after the pre-test intervention. During sociability, the frequency and time spent with conspecific was measured to reflect social interest. The social preference index (SI) was calculated as follows:

$$SI = t_{mouse}/[t_{mouse} + t_{container}] \times 100$$

where $t_{mouse}$ was the percentage of time spent sniffing the container with the stimulus mouse and $t_{container}$ was the percentage of time spent with sniffing the empty container. A value higher than 50 reflects social interest.

At the end of these phases, the animals were put back to their homecages, then 2.5 h as well as 24 h later they were put back into the same test box for testing social discrimination (SDT). SDT is based on the innate preference for novelty of mice. The experimental setting was similar to that of the sociability test: in an empty white plastic box two wired cages with weights on top were placed. Under each of them, one conspecific was placed. One was the same as during sociability (called juvenile 1, 'J1' mouse), thus, J1 was already known to the experimental mice. The other one was an unknown (called 'J2' and 'J3') mouse. The position of the J1 mouse was interchanged compared to previous test to avoid place preference. The experimental animal is expected to spend more time with an unknown conspecific than with an already familiar one. The experimental animal could freely behave for 5 min.

The frequency and time spent with each conspecific were measured. Any other type of behaviour was labelled as 'other'. Social discrimination index (SD) was calculated based on this equation:

$$SD = [t_{mouse1} - t_{mouse2 \text{ or } 3}]/[t_{mouse1} + t_{mouse2 \text{ or } 3}] \times 100$$

A value higher than 0 reflects intact social memory. In each group one mice should have been left out from this analysis as they did not sniff the original stimulus mice at all, thus, could hardly remember them.

The test apparatuses were cleaned with 20% ethanol between animals.

### 4.2.2. Social Interaction Test (SIT)

SIT investigates anxiety-driven social behaviours. The test was conducted in a transparent plexiglass cage (35 cm $\times$ 20 cm $\times$ 25 cm) with bedding on the bottom. The day before, the experiment animals were habituated to the test arena one-by-one for $2 \times 10$ min, 4 h apart. During the test, two identically treated but unfamiliar mice roughly having the same age and bodyweight, were put in the already familiar arena. They could freely interact for 10 min in normal light. Friendly (e.g., sniffing), aggressive (e.g., biting, aggressive dominance) and defensive (e.g., running away) behaviours of both animals were analysed (frequencies and time spent). A Friendly Social index (FI) [23] was calculated based on the following equation:

$$FI = t_{friendly \text{ social}}/[t_{friendly \text{ social}} + t_{aggressive} + t_{defensive}] \times 100$$

### 4.2.3. Resident–Intruder Test (RIT)

RIT measures territorial aggression. An unfamiliar, smaller, but sexually mature conspecific was put into the home cage of the test animals for 10 min under infrared light. Only the behaviour of the test animal was analysed with the same parameters as in SIT.

### 4.3. Stress Hormone Measurements

Blood samples were collected 30 min after manipulation, or at the time of behavioural testing on other days. The blood was kept on ice until centrifugation at 4 °C and $2500 \times g$ for 30 min. The serum was kept at $-20$ °C until processed. ACTH levels were measured by radioimmunoassay in 50 µL unextracted serum in duplicates using a specific antibody developed at the Institute of Experimental Medicine (Budapest, Hungary) [46]. The intra-assay coefficient of variation was 7.5% and all samples from the experiment were measured in one session. The ACTH levels of all the animals took part in the behavioural test battery were measured.

*4.4. Statistical Analysis*

The data were analysed by the StatSoft 13.4 software (StatSoft, Inc., TIBCO, Palo Alto, CA, USA) using one-way ANOVA. For the SI and DI parameters, single-sample *t*-test compared to 50% or 0%, respectively, was used for each group separately. The comparison of side during the object habituation phase (left vs. right), social preferences in sociability (mouse vs. object) as well as social memory test were analysed by repeated-measures ANOVA. Posthoc comparison, where appropriate, was done using the Tukey HSD test. Data were expressed as mean $\pm$ SEM and $p < 0.05$ was considered statistically significant.

## 5. Conclusions

All in all, intraperitoneal injections with or without isoflurane anaesthesia might be used without violating the 3R principle and influencing the desired outcome of social behavioural experiments in male C57BL/6J mice. However, we cannot rule out the possibility that repeated isoflurane inhalation might be harmful not only to the animals [2,29], but also to the observers. Indeed, in humans two isoflurane inhalations can lead to alteration in liver enzymes [47]. Moreover, our present results also suggested some reduction in social interest 40 min after isoflurane inhalation plus saline injection. Thus, in the hand of a trained person, we recommend using intraperitoneal injection without initial inhalation anaesthesia in male C57BL/6J mice, as there was no benefit gained from it.

**Author Contributions:** Conceptualization, D.Z.; methodology, F.P. and B.T.; formal analysis, F.P., B.T. and D.Z.; investigation, F.P. and B.T.; resources, D.Z.; data curation, F.P., B.T. and D.Z.; writing—original draft preparation, F.P., B.T. and D.Z.; visualization, F.P., B.T. and D.Z.; project administration, D.Z.; funding acquisition, D.Z. All authors have read and agreed to the published version of the manuscript.

**Funding:** This study was supported by the Hungarian Brain Research Program 3.0, the National Research Development and Innovation Office of Hungary (grant numbers K141934, K138763 and K120311), as well as the Thematic Excellence Program 2021 Health Sub-program of the Ministry for Innovation and Technology in Hungary (within the framework of the TKP2021-EGA-16 project of the Pécs University).

**Institutional Review Board Statement:** All tests were approved by the local committee of animal health and care (PEI/001/33-4/2013, PE/EA/254-7/2019) and performed according to the European Communities Council Directive recommendations for the care and use of laboratory animals (2010/63/EU).

**Informed Consent Statement:** Not applicable.

**Data Availability Statement:** Data are available upon request.

**Conflicts of Interest:** The authors declare no conflict of interest. The funders had no role in the design of the study; in the collection, analyses, or interpretation of data; in the writing of the manuscript; or in the decision to publish the results.

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
