# Peer review of "Pre-Test Manipulation by Intraperitoneal Saline Injection with or without Isoflurane Pre-Treatment Does Not Influence the Outcome of Social Test in Male Mice"

_stresses, doi:10.3390/stresses3010025_

Round 1

Reviewer 1 Report

In the manuscript entitled “Pretest manipulation with isoflurane or intraperitoneal saline injection does not influence the outcome of social test in mice”, the authors aimed to test whether intraperitoneal injection alone or in combination with isoflurane anaesthesia is a preferable treatment method 30 min before a social test. Methodology applied is appropriate and the conclusion is in line with the stated goal.

What remains my main complaint is the inconsistency in the explanation of the treatment of the third experimental group. Namely, in the abstract (line 12 and 13), the authors state that they are examining the impact of intraperitoneal injection, with or without prior application of isoflurane. Furthermore, in the last part of the introduction (line 82 and 83), it is stated that the influence of intraperitoneal injection OR anesthetization with isoflurane is examined, which suggests that these are separate treatments, i.e. that isoflurane anesthesia is not followed by intraperitoneal injection. Such a suggestion is further confirmed by the scheme of the experiment (Fig1). In line 116, it is stated that one group of animals was pretreated with isoflurane, which suggests that treatment was performed afterwards. However, it remains unclear what this treatment is. Furthermore, in the Material and methods section, line 180, the third experimental group is described, but based on that description it can only be concluded that the animals were anesthetized with isoflurane but not that they were injected intraperitoneally afterwards. Finally, in the conclusion, line 257, it is concluded on the application of intraperitoneal injection, with OR without prior anesthesia.

Although the results are such that it can be concluded that the hypothesis could have been tested regardless of whether the third group was injected intraperitoneally or not, the design of the experiment had to include anesthesia with isoflurane as a pretreatment to the intraperitoneal injection. This is indeed how the authors designed the experiment, but it is necessary that it be clearly visible in every part of the manuscript, in order to avoid misinterpretation. For this purpose, I suggest that the authors standardize the description of the manipulations with the third group of animals, as well as to consider correcting the name of the third experimental group (including the data in Fig. 1), for example Isoflurane+Saline.

In addition to the above-mentioned major suggestions, some minor ones follow in the following text.

Abstract, Line 14 and 15: it is not clearly stated what are the second and third experimental group. That part of the sentence should be rephrased (Something like: Intra-peritoneal saline treated groups with or without previous short isoflurane inhalation).

Line 85: there is a misspelling in the word interction in the explanation of the Figure 1. Also, I believe that word intul is supposed to be until.

Line 95: there is a misspelling in the word grousp in the explanation of the Figure 2.

Line 104: there is a misspelling in the words indeces and grousp in the explanation of the Figure 3.

Line 119: there is a misspelling in the words tehre and teh in the explanation of the Figure 4.

Line 120: the authors stated that the friendly behavior increased during the RIT. That could be truth if the results of control groups in SIT and RIT were statistically compared. Were those results compared? If yes, please state that.

Line 172: ad libitum should be written in cursive

Word behaviour should be written at the same manner throughout the manuscript, for example behaviour/behavior is written differently in lines 173 and 174. Check it throughout the text.

Author Response

Reviewer 1.

Thank you very much for the positive evaluation and helpful suggestions. We made changes according to your suggestions and the modifications are highlighted in the manuscript. The point-by-point answer can be found bellow.

What remains my main complaint is the inconsistency in the explanation of the treatment of the third experimental group. Namely, in the abstract (line 12 and 13), the authors state that they are examining the impact of intraperitoneal injection, with or without prior application of isoflurane. Furthermore, in the last part of the introduction (line 82 and 83), it is stated that the influence of intraperitoneal injection OR anesthetization with isoflurane is examined, which suggests that these are separate treatments, i.e. that isoflurane anesthesia is not followed by intraperitoneal injection. Such a suggestion is further confirmed by the scheme of the experiment (Fig1). In line 116, it is stated that one group of animals was pretreated with isoflurane, which suggests that treatment was performed afterwards. However, it remains unclear what this treatment is. Furthermore, in the Material and methods section, line 180, the third experimental group is described, but based on that description it can only be concluded that the animals were anesthetized with isoflurane but not that they were injected intraperitoneally afterwards. Finally, in the conclusion, line 257, it is concluded on the application of intraperitoneal injection, with OR without prior anesthesia. Although the results are such that it can be concluded that the hypothesis could have been tested regardless of whether the third group was injected intraperitoneally or not, the design of the experiment had to include anesthesia with isoflurane as a pretreatment to the intraperitoneal injection. This is indeed how the authors designed the experiment, but it is necessary that it be clearly visible in every part of the manuscript, in order to avoid misinterpretation. For this purpose, I suggest that the authors standardize the description of the manipulations with the third group of animals, as well as to consider correcting the name of the third experimental group (including the data in Fig. 1), for example Isoflurane+Saline.

Answer: Thank you very much for the clarification. We changed the text as well as figures even the title accordingly.

In addition to the above-mentioned major suggestions, some minor ones follow in the following text.

Abstract, Line 14 and 15: it is not clearly stated what are the second and third experimental group. That part of the sentence should be rephrased (Something like: Intra-peritoneal saline treated groups with or without previous short isoflurane inhalation).

Answer: The sentences was modified “one control group and intraperitoneal saline treated groups with or without short isoflurane inhalation”.

Line 85: there is a misspelling in the word interction in the explanation of the Figure 1. Also, I believe that word intul is supposed to be until.

Answer: Corrected according to suggestions.

Line 95: there is a misspelling in the word grousp in the explanation of the Figure 2.

Answer: Corrected according to the suggestion.

Line 104: there is a misspelling in the words indeces and grousp in the explanation of the Figure 3.

Answer: Corrected according to suggestions.

Line 119: there is a misspelling in the words tehre and teh in the explanation of the Figure 4.

Answer: Corrected according to suggestions.

Line 120: the authors stated that the friendly behavior increased during the RIT. That could be truth if the results of control groups in SIT and RIT were statistically compared. Were those results compared? If yes, please state that.

Answer: They were compared by repeated measure ANOVA as stated in the text: (repeated measure ANOVA on FIs: F(1,25)=4.30, p=0.05). However, it was not clear that the comparison was between SIT and RIT. Now we clarified it.

Line 172: ad libitum should be written in cursive

Answer: Corrected according to the suggestion.

Word behaviour should be written at the same manner throughout the manuscript, for example behaviour/behavior is written differently in lines 173 and 174. Check it throughout the text.

Answer: We corrected all words according to British English, as behaviour.

Reviewer 2 Report

0_Overview

The authors examined whether pretreatment with two possible stress factors, i.e., intraperitoneal injection and isoflurane anesthesia, could influence experimental results of social behaviors and stress hormone levels in mice. It was shown that both stress factors had little effect on conventional parameters of social tests and serum adrenocorticotropic hormone (ACTH) levels in male C57BL/6J mice 30 min after stress induction. Based on these results, the authors concluded that not only intraperitoneal injection but also isoflurane anesthesia might hardly affect behavioral experiments as stressors in mice.

The manuscript as a whole is well-written and clear, but some points should be properly addressed before publication. Comments to the authors are as follows.

1_Major comments

1a_The results of this study are very interesting and surprising, especially mild stress effect of intraperitoneal injection in mice is. Many readers related laboratory animal science probably hope more precise explanation of methods how an experimenter gave injection to a subject animal, for example, needle diameter and syringe volume used, whether animal retention was performed or not, and so on. Please describe these in your revised manuscript.

1b_ 5. Conclusions: Regarding the first sentence (Page 8, Lines 257─259), it should be better to change “the behavioral experiments” into “the social behavioral experiments”, since the findings in this study were obtained from only several kinds of social behavioral tests. It is unclear, in fact, whether this suggestion can be applied to other behavioral tests such as elevated plus maze test, light-dark preference test, acoustic startle reflex test, and so on. Moreover, from a different point of view, this sentence and the fifth sentence (Pages 8─9, Lines 263─265) are a bit over-speculation, since the authors simply assessed social behaviors and serum ACTH revels only using male C57BL/6J mice. There is yet no information whether it is possible to reproduce this study’s phenomena in other species, other strains, and females to date. Therefore, I recommend rewriting these sentences as follows.

─ (Page 8, Lines 257─259) ─> “All in all, intraperitoneal injections with or without isoflurane anaesthesia might be used without violating the 3R principle and influencing the desired outcome of the social behavioural experiments in male C57BL/6J mice.”

─ (Pages 8─9, Lines 263─265) ─> “Thus, in the hand of a trained person, we recommend using intraperitoneal injection without initial inhalation anaesthesia in male C57BL/6J mice, as there was no gained benefit from it.”

Finally, in line with these points of view, it should be better to discuss such limitations in the Discussion section.

2_Minor comments

2a_ It seems to me that the first paragraph in the Introduction section is a bit long. I would suggest breaking it two paragraphs and starting the second paragraph from the sentence “As pain induce stress in the animals… (Page 1, Line 35)”.

2b_ Page 1, Line 28: “understading” ─> “understanding”

2c_I would suggest rewriting the last paragraph of the Introduction section (Page 2, Lines 8284) on the basis of the following example: “On the basis of all above mentioned findings, it is considered that there is insufficient information to evaluate whether intraperitoneal injection and/or isoflurane are stressors to influence experimental results of social behaviors in laboratory mice. Therefore, here we aimed to study whether intraperitoneal injection or isoflurane anesthesia applied 30 min before a social test has any effect on the measured behavioral outcome and plasma ACTH levels (for the details of the experiments see Figure 1).”

2d_ It would be better to move the second paragraph of the Results section regarding serum ACTH levels (Page 4, Lines 9295) to the last paragraph in this section, together with rearrangement of related parts in Table 1 and the order of Figures (Figure 2A ─> the last Figure).

2e_ Page 6, Line 138: “Luckily” It seems to me that this word is a bit inappropriate. Please rephrase or delete it.

2f_ 4.1. Animals: What did you use for bedding in mouse cages? Please describe it in your revised manuscript.

2g_ Page 7, Lines 185─186: “day 10. sacrifice for blood sampling 30 min after interventions” Please explain more precisely how the euthanasia of animals was performed for blood sampling, for example, decapitation or cardiopuncture (or other methods), anesthetized or not (if anesthetized, describe what types of anesthetics you used), and so on.

2h_ Page 8, Lines 221─223: “Some of the mice should have been left out from this analysis as they did not sniff the original stimulus mice enough to be able to remember them.” It is a little difficult to understand what this sentence means. Please rephrase it more clearly.

2i_ I think that there is a discrepancy between the following descriptions in the Discussion section and the Conclusions section.

The Discussion section (Page 6, Lines 164─166) “Indeed, in accordance with the supposed pain- and thereby stress-reducing effect of isoflurane anaesthesia [28], mice were more friendly 30 min after its inhalation (Fig.4A).”

The Conclusions section (Page 9, Lines 265─267) “Our results are in accordance with previous reports showing that the use of isoflurane induction before a stressful event significantly increased, and not reduced, both behavioral and neuromolecular signs of stress [48].”

Please explain properly.

2j_ Table 1: It should be better to place captions above the table.

Author Response

Reviewer 2.

Thank you very much for the positive evaluation and helpful suggestions. We made changes accordingly and highlighted them in the manuscript. The point-by-point answers can be found bellow.

1_Major comments

 1a_The results of this study are very interesting and surprising, especially mild stress effect of intraperitoneal injection in mice is. Many readers related laboratory animal science probably hope more precise explanation of methods how an experimenter gave injection to a subject animal, for example, needle diameter and syringe volume used, whether animal retention was performed or not, and so on. Please describe these in your revised manuscript.

Answer: A more detailed description of the ip injection was provided. “(26G needle, 1ml syringe, <1 min immobilised by the skin of the back)”

 1b_ 5. Conclusions: Regarding the first sentence (Page 8, Lines 257─259), it should be better to change “the behavioral experiments” into “the social behavioral experiments”, since the findings in this study were obtained from only several kinds of social behavioral tests. It is unclear, in fact, whether this suggestion can be applied to other behavioral tests such as elevated plus maze test, light-dark preference test, acoustic startle reflex test, and so on. Moreover, from a different point of view, this sentence and the fifth sentence (Pages 8─9, Lines 263─265) are a bit over-speculation, since the authors simply assessed social behaviors and serum ACTH revels only using male C57BL/6J mice. There is yet no information whether it is possible to reproduce this study’s phenomena in other species, other strains, and females to date. Therefore, I recommend rewriting these sentences as follows.

─ (Page 8, Lines 257─259) ─> “All in all, intraperitoneal injections with or without isoflurane anaesthesia might be used without violating the 3R principle and influencing the desired outcome of the social behavioural experiments in male C57BL/6J mice.”

─ (Pages 8─9, Lines 263─265) ─> “Thus, in the hand of a trained person, we recommend using intraperitoneal injection without initial inhalation anaesthesia in male C57BL/6J mice, as there was no gained benefit from it.”

Answer: Thank you very much for the clarification. We made changes accordingly.

Finally, in line with these points of view, it should be better to discuss such limitations in the Discussion section.

Answer: Limitation was added to discussion. “There is yet no information whether it is possible to reproduce this study’s phenomena in other species, other strains, and females to date. A further limitation of our study is its focus on social behaviour.”

 2_Minor comments

 2a_ It seems to me that the first paragraph in the Introduction section is a bit long. I would suggest breaking it two paragraphs and starting the second paragraph from the sentence “As pain induce stress in the animals… (Page 1, Line 35)”.

Answer: We made changes accordingly.

2b_ Page 1, Line 28: “understading” ─> “understanding”

 Answer: We made changes accordingly.

2c_I would suggest rewriting the last paragraph of the Introduction section (Page 2, Lines 82─84) on the basis of the following example: “On the basis of all above mentioned findings, it is considered that there is insufficient information to evaluate whether intraperitoneal injection and/or isoflurane are stressors to influence experimental results of social behaviors in laboratory mice. Therefore, here we aimed to study whether intraperitoneal injection or isoflurane anesthesia applied 30 min before a social test has any effect on the measured behavioral outcome and plasma ACTH levels (for the details of the experiments see Figure 1).”

Answer: We made changes accordingly. 

2d_ It would be better to move the second paragraph of the Results section regarding serum ACTH levels (Page 4, Lines 92─95) to the last paragraph in this section, together with rearrangement of related parts in Table 1 and the order of Figures (Figure 2A ─> the last Figure).

 Answer: After long consideration we would like to keep the placement of ACTH data as it is, as they are not social parameters, more belong to general parameters like BW or locomotion. Especially, logical rearrangement of figures would be rather hard.

2e_ Page 6, Line 138: “Luckily” It seems to me that this word is a bit inappropriate. Please rephrase or delete it.

Answer: The word was deleted.

 2f_ 4.1. Animals: What did you use for bedding in mouse cages? Please describe it in your revised manuscript.

Answer: The information was added.” corn cob bedding”

 2g_ Page 7, Lines 185─186: “day 10. sacrifice for blood sampling 30 min after interventions” Please explain more precisely how the euthanasia of animals was performed for blood sampling, for example, decapitation or cardiopuncture (or other methods), anesthetized or not (if anesthetized, describe what types of anesthetics you used), and so on.

 Answer: The following clarification was added: by decapitation without further anaesthesia”

2h_ Page 8, Lines 221─223: “Some of the mice should have been left out from this analysis as they did not sniff the original stimulus mice enough to be able to remember them.” It is a little difficult to understand what this sentence means. Please rephrase it more clearly.

 Answer: The sentence was modified: “In each group one mice should have been left out from this analysis as they did not sniff the original stimulus mice at all, thus, hardly could remember them.”

2i_ I think that there is a discrepancy between the following descriptions in the Discussion section and the Conclusions section.

The Discussion section (Page 6, Lines 164─166) “Indeed, in accordance with the supposed pain- and thereby stress-reducing effect of isoflurane anaesthesia [28], mice were more friendly 30 min after its inhalation (Fig.4A).”

The Conclusions section (Page 9, Lines 265─267) “Our results are in accordance with previous reports showing that the use of isoflurane induction before a stressful event significantly increased, and not reduced, both behavioral and neuromolecular signs of stress [48].”

Please explain properly.

Answer: The friendly index (Fig 4A) indeed was elevated, however, sociability in general (Fig 3A) was decreased, therefore we concluded that signs of stress increased. To avoid misleading statement, we deleted the last sentence of the conclusion.

 2j_ Table 1: It should be better to place captions above the table.

Answer: Changes were made accordingly.